# Quantifying the Impact of Light Pollution on Sea Turtle Nesting Using Ground-Based Imagery

**James Vandersteen [1,*], Salit Kark [1], Karina Sorrell [2] and Noam Levin [3,4]**

[1] The Biodiversity Research Group, School of Biological Sciences, Centre for Biodiversity and Conservation Science, The University of Queensland, St Lucia, QLD 4072, Australia; s.kark@uq.edu.au

[2] School of Geography, The University of Melbourne, Parkville, VIC 3010, Australia; karina.sorrell@unimelb.edu.au

[3] Department of Geography, The Hebrew University of Jerusalem, Jerusalem 91905, Israel; noamlevin@mail.huji.ac.il or n.levin@uq.edu.au

[4] School of Earth and Environmental Sciences, The University of Queensland, St Lucia, QLD 4072, Australia

* Correspondence: james.vandersteen@sydney.edu.au

**Abstract:** Remote sensing of anthropogenic light has substantial potential to quantify light pollution levels and understand its impact on a wide range of taxa. Currently, the use of space-borne night-time sensors for measuring the actual light pollution that animals experience is limited. This is because most night-time satellite imagery and space-borne sensors measure the light that is emitted or reflected upwards, rather than horizontally, which is often the light that is primarily perceived by animals. Therefore, there is an important need for developing and testing ground-based remote sensing techniques and methods. In this study, we aimed to address this gap by examining the potential of ground photography to quantify the actual light pollution perceived by animals, using sea turtles as a case study. We conducted detailed ground measurements of night-time brightness around the coast of Heron Island, a coral cay in the southern Great Barrier Reef of Australia, and an important sea turtle rookery, using a calibrated DSLR Canon camera with an 8 mm fish-eye lens. The resulting hemispheric photographs were processed using the newly developed Sky Quality Camera (SQC) software to extract brightness metrics. Furthermore, we quantified the factors determining the spatial and temporal variation in night-time brightness as a function of environmental factors (e.g., moon light, cloud cover, and land cover) and anthropogenic features (e.g., artificial light sources and built-up areas). We found that over 80% of the variation in night-time brightness was explained by the percentage of the moon illuminated, moon altitude, as well as cloud cover. Anthropogenic and geographic factors (e.g., artificial lighting and the percentage of visible sky) were especially important in explaining the remaining variation in measured brightness under moonless conditions. Night-time brightness variables, land cover, and rock presence together explained over 60% of the variation in sea turtle nest locations along the coastline of Heron Island, with more nests found in areas of lower light pollution. The methods we developed enabled us to overcome the limitations of commonly used ground/space borne remote sensing techniques, which are not well suited for measuring the light pollution to which animals are exposed. The findings of this study demonstrate the applicability of ground-based remote sensing techniques in accurately and efficiently measuring night-time brightness to enhance our understanding of ecological light pollution.

**Keywords:** ecological light pollution; hemispheric photography; Sky Quality Camera; moon; clouds; Great Barrier Reef

## 1. Introduction

Artificial light is increasingly recognised as a form of environmental pollution. Ecological light pollution refers to artificial light that alters the natural light regime and adversely affects wildlife [1]. The recognition of light as a form of pollution is relatively new, and was first identified by astronomers [2]. Whilst anthropogenic impacts such as global warming, land clearing, and more tangible forms of pollution are relatively well studied, the ecological impacts of light pollution are less known [3]. However, much of the literature suggests that light pollution is detrimental to an array of wildlife [3–7]. Though the negative consequences of light pollution are likely far reaching and complex, a critical aspect is the direct and indirect mortality of individuals. Such events resulting from light pollution have been recorded in species of insects, birds, fish, and sea turtles [8–11].

Sea turtles represent a group of species for which the negative effects of light pollution have been most studied, using both space-borne [12,13] and ground-based [14–16] remote sensing techniques. Complex relationships have been found between female sea turtle nest site selection and artificial light. It has been widely concluded that females display a preference for nesting in low light/natural conditions [16–19]. The degree of night-time brightness in any given area is however, not limited to anthropogenic features. Moon presence and cloud cover are important environmental factors that also naturally affect night-time brightness [20–23]. In-fact, previous studies have distinguished these factors and demonstrated sea turtles' abilities to perceive them with regards to light pollution [23–26]. Specifically, artificial light is less conspicuous under gibbous moon (greater than half-moon) and/or clear sky conditions because the ambient night-time light is already relatively bright due to the moon and stars, and their non-obstruction by clouds. Conversely, artificial light becomes more prominent under a crescent moon (less than half-moon) and/or cloudy conditions for the opposite reasons, and because of the reflective nature of clouds exacerbating light pollution in areas with artificial lights, and darkening the skies in pristine areas [27–29]. Therefore, to mitigate the impacts of light pollution on sea turtles, and indeed other ground dwelling species, a reliable method for measuring brightness is needed, a method which can also take into consideration the effect of moon illumination and cloud cover.

Remote sensing can be used for ecological research and may be especially beneficial when employed in ecological studies situated in large and/or isolated areas, and for collecting digital and quantitative data which can be visually presented [30–32]. Recently, air and space borne remote sensing (utilising satellites, aircrafts, and drones) has opened many avenues for studying light pollution [33]. These tools can provide a comprehensive view of anthropogenic light over large spatial and temporal scales, making global observations quick and convenient [34,35]. However, aerial and space-borne sensors are less suited for ecological purposes because they mostly measure artificial light emitted upwards [36] and not light emitted horizontally that is perceived by many ground-dwelling species [33]. Furthermore, most freely available space borne night-time images are only available at course spatial resolutions [37] (VIIRS/DNB at 750 m, and DMSP/OLS at ~3 km; [35]) and are thus not suited to study local sources of light pollution. Local sources of light pollution can instead be examined and quantified using ground-based remote sensing. Instruments for such purposes include basic night sky brightness photometers such as Sky Quality Meters (SQM) [38] or the TESS-W photometer [39]. Digital Single Lens Reflex (DSLR) cameras equipped with fisheye lenses for wide angle hemispheric photography (i.e., 180° field of view) are a newly available technology which has not been widely used in scientific research. These cameras and the accompanying Sky Quality Camera (SQC) software provide a good compromise between ease-of-use and obtainable information for the study of light pollution and the associated ecological impacts [40–44].

As human development continues to expand, the amount of light pollution worldwide is expected to increase in both area and radiance [45]. Thus, parameters for quantifying artificial light such as brightness metric guidelines and brightness indices [46] will need to be developed and implemented rather than simply aiming to minimise light pollution alone. With the recent development of the SQC software, which enables analysis of hemispheric night-time photographs acquired using a DSLR camera (calibrated to work with this software by developer Andrej Mohar), we sought to

demonstrate its applicability for studying ecological light pollution. Specifically, our aim was to refine the methodologies for and promote the application of such technology by examining how brightness metrics varied as a function of several environmental and anthropogenic variables within an ecologically sensitive area. We predicted that brightness would increase with greater percentages of the moon illuminated, greater cloud cover, and closer proximity to artificial lighting sources. Conversely, we expected that brightness would decrease as a function of light obstruction by vegetation, resulting in a brighter seaward horizon than landward horizon. We were also interested in knowing to what extent environmental and anthropogenic factors (i.e., light pollution) contribute to explaining the nesting locations of green turtles (*Chelonia mydas*) and loggerhead turtles (*Caretta caretta*). We predicted that the presence of rock outcrops on the beach and high levels of night-time brightness as a function of light pollution would result in fewer sea turtles nesting. This study aimed to conduct a comprehensive analysis of the spatial patterns of light pollution comparing several brightness metrics, using a ground based camera and the SQC software, to quantify the ecological impacts of light pollution, using sea turtles as a model species.

## 2. Methods

### 2.1. Study Site

We conducted fieldwork on Heron Island (23.4423°S, 151.9148°E), a coral cay located in the southern Great Barrier Reef approximately 80 km east of mainland Australia. Heron Island was selected due to its small area and perimeter (16.8 ha and 1.8 km), and concentrated human development (The Heron Island Resort and The University of Queensland Heron Island Research Station; Figure 1). This facilitated the investigation of light pollution, and its role in providing habitat for sea turtle nesting which has been annually monitored and recorded for approximately 50 years [47].

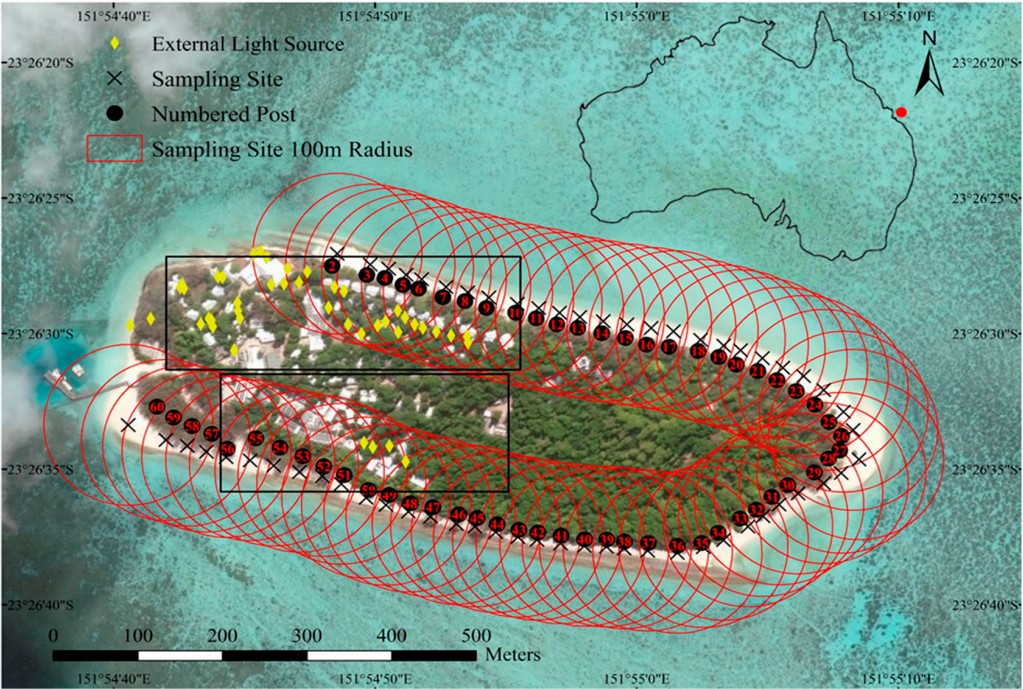

**Figure 1.** Satellite image of Heron Island showing the locations of the 59 sampling sites (black crosses), the corresponding 100 metre radius around each site (red circles), and the external lighting sources (yellow diamonds). The University of Queensland Research Station buildings in the south-west are shown in the bottom black rectangle and the Heron Island Resort buildings in the north-west are shown in the top black rectangle. Inset map shows Heron Island's location (red dot) relative to mainland Australia.

*2.2. Fieldwork*

### 2.2.1. Sampling Site Selection

The Heron Island coastline is sectioned with 64 numbered posts to assist ongoing data collection of sea turtle population trends, dynamics, and nest monitoring. Photographic sampling sites were selected to correspond with 59 of the 64 pre-existing numbered posts (Figure 1) and were located five meters below the king tide mark i.e., the highest tide mark identified based on sediment differentiation and debris (Figure 2). Selection of sampling sites based on the posts allowed for coherence between the photographic samples and the sea turtle data.

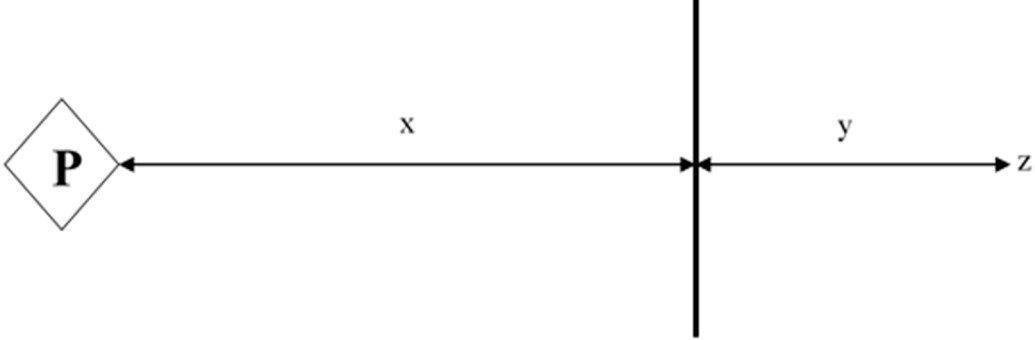

**Figure 2.** Schematic layout of the method used to select a sampling site—where the diamond denoted with a 'P' represents the post (landward), 'x' represents the distance from the post to the king tide mark, the solid vertical line represents the king tide mark, 'y' represents the 5-m distance below the king tide mark, and 'z' represents the sampling site (seaward).

### 2.2.2. Equipment

For the photographic sampling we used a Canon 6D EOS DSLR Camera with an attached Sigma Lens 8 mm EX DG Circular Fisheye. The camera, lens, and SQC software which has been suggested to be useful for quantifying ecological light pollution [42], were specifically calibrated by Euromix (Slovenia) for such purposes. The camera was placed on a leveled tripod approximately one meter above the ground. The three-dimensional placement of the camera (i.e., in line with corresponding post, distance below the king tide mark, and height from the ground) was chosen to meet the best compromise between ecological relevance for sea turtles and comprehensive measurements of brightness.

### 2.2.3. Sampling Sessions

Photographic sampling was undertaken in two blocks; 29 April 2018–16 May 2018 and 13 June 2018–30 June 2018, to coincide with the transition between a full moon and new moon (Table S1). To control for the effects of moon presence on brightness, half of the sampling sessions were conducted under moonlit conditions and the other half under moonless conditions. Three photographic samples were taken at each site within a sampling session. The first photograph at each site faced the zenith (i.e., facing directly upwards towards the sky; hereon named 'zenith photograph'). The other two photographs at each site were taken with the lens horizontally level and aligned with the corresponding post, one facing the seaward direction and one facing the landward direction (hereon collectively named 'horizontal photographs' and separately named 'horizontal seaward photograph' and 'horizontal landward photograph'). Each photograph had an ISO of 1600, aperture of 3.5, and exposure time between 5–90 seconds as recommended by SQC guidelines and readings of image quality (i.e., shorter exposure under moonlit conditions, longer exposure under moonless conditions). We conducted sampling regardless of cloud cover, however, cloud cover was quantified via the SQC software using the zenith photographs.

## *2.3. Mapping and Image Analysis*

### 2.3.1. Mapping of Light Sources

To estimate proxies for light pollution we mapped all accessible external light sources and buildings on Heron Island to calculate the combined numbers of each within a 100metre radius of each sampling site (Figure 1).

### 2.3.2. Classification of Land Cover

We were interested in whether brightness was explained as a function of artificial light emission and its obstruction by vegetation. To explore this, we used Envi to conduct a supervised classification (Support Vector Machine) of a WorldView 3 satellite image of Heron Island (acquired on 14 November 2015) to the following four classes: water; vegetation; buildings; ground (Figure 3). Using this classified image, we calculated the percentage cover of buildings and of vegetation within a 100metre radius of each sampling site.

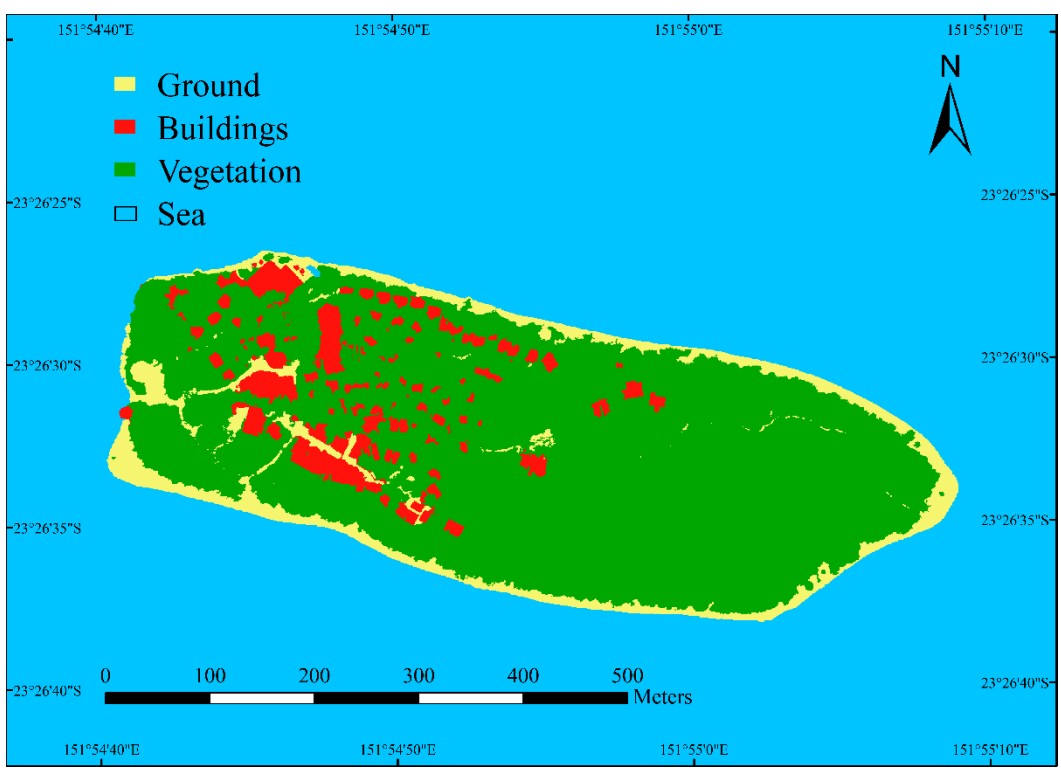

**Figure 3.** Classified satellite image of Heron Island.

### 2.3.3. Measurement of the Percentage of Visible Sky

Hemispheric photographs are commonly used to quantify plant canopy [48]. We wanted to quantify the percentage of visible sky based on one landward hemispheric photograph (taken during the day-time) at each sampling site using a supervised classification tool within ImageJ software: Trainable Weka Segmentation tool (Figure 4) [49]. We expected that under night-time conditions the celestial lit sky and artificial light sources would be brighter than the vegetation obstructing both these light sources.

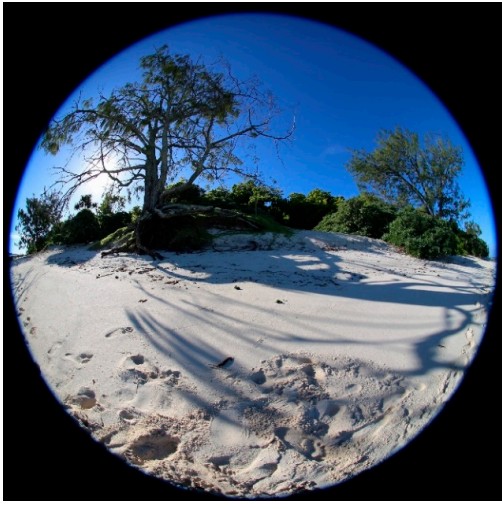 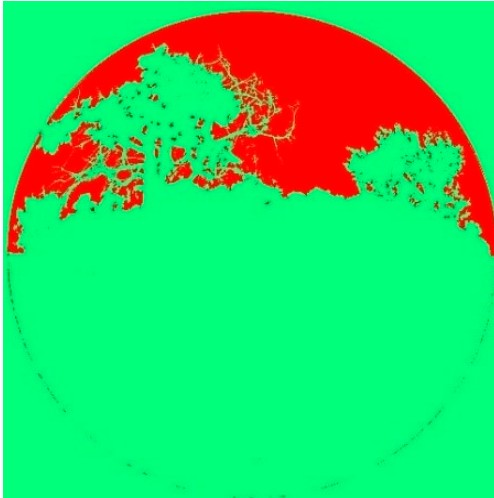

**Figure 4.** Example of a supervised classification using ImageJ Trainable Weka Segmentation. This figure depicts a classified image (**right**) of the photograph (**left**) taken 29 June 2018 (during the day-time) at sampling site 46 Heron Island, where the sky is displayed in red and everything else (i.e., vegetation and sand) in green.

### 2.3.4. Measurement of Beach Features

As sea turtle nesting is not solely influenced by brightness we also calculated beach width and rock outcrop presence, two commonly recognised beach features that influence sea turtle nest site selection [18,19]. The average beach width at each sampling site was calculated by averaging the measured high tide (30 January 2018) and low tide (7 June 2018) beach width using two Planet Labs satellite images of Heron Island. Satellite imagery was also used to record the presence/absence of a rock outcrops at each sampling site.

### 2.3.5. Extraction of Brightness Metrics Using SQC Software

We used Sky Quality Camera (SQC) version 1.8.0 software for photographic calibration and analysis to extract brightness metrics. Measurements of brightness were provided in units of Magnitude per Square Second of Arc (V mag/arcsec$^2$), where the brightness in magnitudes is spread over a square arcsecond of the sky, with lower values signifying higher brightness (Figure 5; Table S2).

Previous studies which have used SQMs or specially developed astronomical cameras to measure brightness, were limited by their ability to differentiate light pollution metrics within specific bounds [14,16]. These limitations are specifically apparent regarding sea turtles who perceive brightness within a hypothetical cone of acceptance (COA) which is confined to 10–30° vertically above the horizon and 180° horizontally wide [8,50]. Therefore, our methods were adapted to facilitate differentiation/calculation of average brightness (V mag/arcsec$^2$) within specifically defined sectors for each photograph, allowing for more ecologically relevant measurements. Within each zenith photograph three sectors were defined:

1. SQM sector—a circular sector from zenith angle 0–30°, defined to represent traditional SQM measurements which are often directed upwards, to assess their relevance for measuring ecological light pollution, specifically in relation to sea turtles, who are unlikely to look upwards (Figure 6(a1)).

2&3. Seaward COA sector & landward COA sector–both sectors 180° horizontally wide and 30° vertically above the horizon, one on the seaward horizon (2) and one on the landward horizon (3) of the image, both defined to represent a sea turtle's COA (Figure 6(a2,a3)).

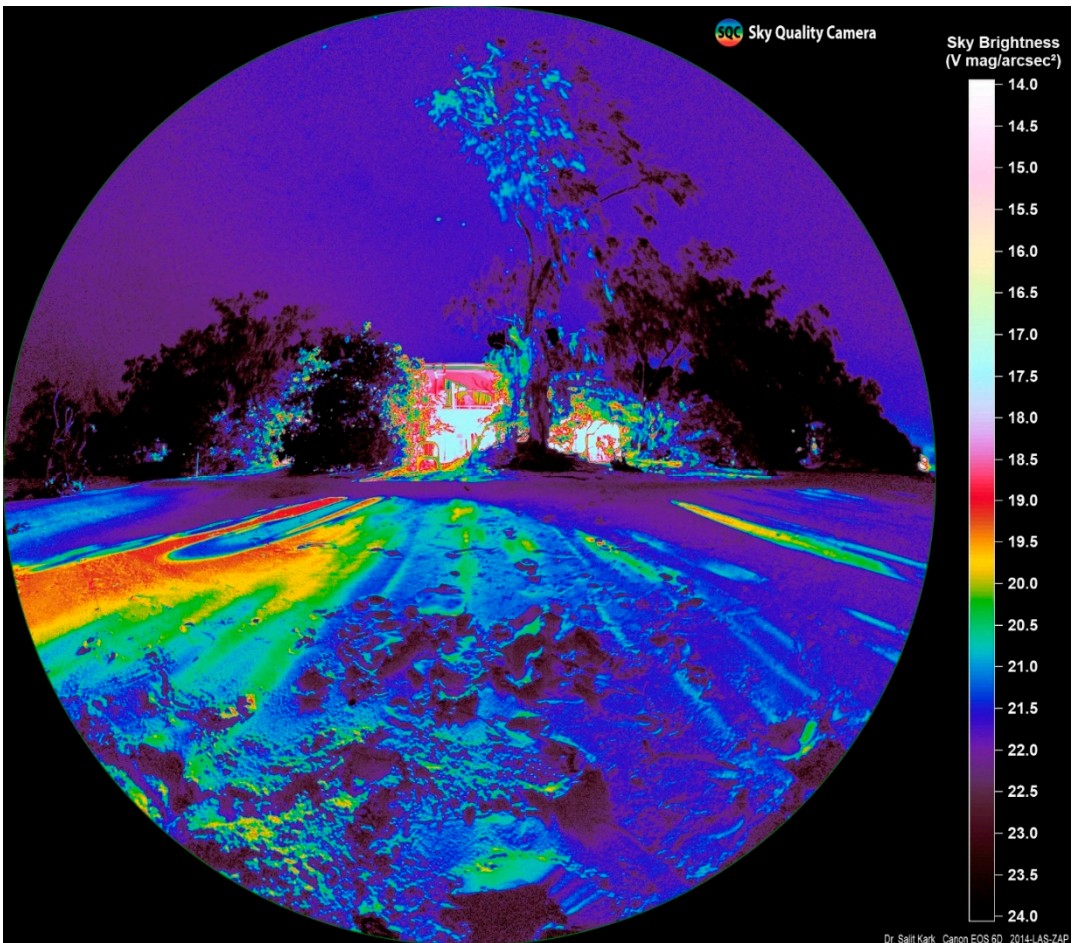

**Figure 5.** Sky Quality Camera calibrated and analysed horizontal landward facing photograph taken on the night of 17 June 2018 at sampling site 5 (facing the resort) Heron Island, Australia (refer to Figure 6b for the raw image).

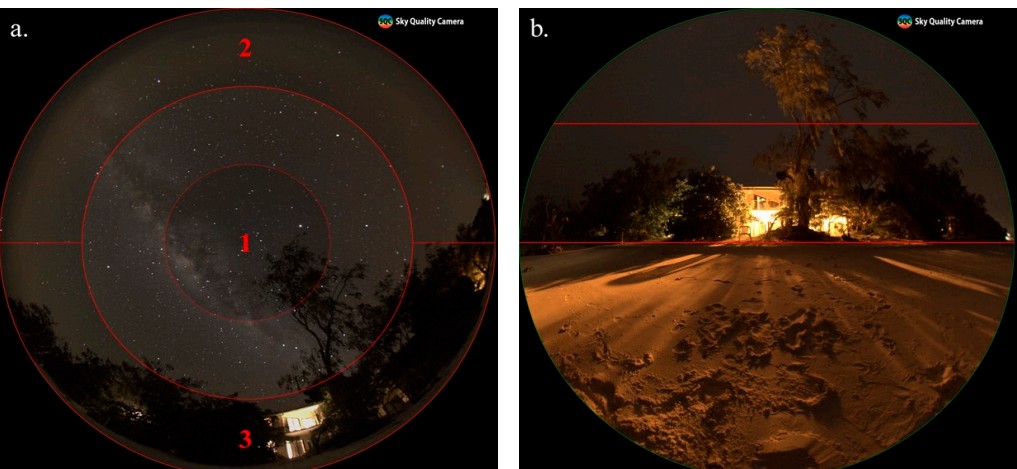

**Figure 6.** (**a**) Zenith photograph taken on the night of 13 May 2018 at sampling site 4 Heron Island, Australia. This image depicts the three sectors classified for sky quality camera brightness analysis: 1. Sky Quality Meter sector, 2. Seaward cone of acceptance sector, 3. Landward cone of acceptance sector. (**b**) Horizontal landward photograph taken on the night of 20 June 2018 at sampling site 4 Heron Island, Australia. This image depicts the COA sector classified for sky quality camera brightness analysis.

A similar sector to that of sectors 2 and 3 was defined for the horizontal photographs to provide another measure of average brightness for the sea turtle's COA (Figure 6b). For each photograph, the SQC software also provided the percentage of the moon illuminated, moon altitude, and the percentage of cloud cover.

### 2.4. Statistical Analysis

R (version 3.5.1) [51] was used for all statistical analyses. Assumptions of homogeneity and normality of residuals were tested for all data and when not met for raw or transformed data, equivalent non-parametric tests were used. A significance level of <0.05 was used for all statistical analyses.

#### 2.4.1. Brightness on Heron Island

An initial analysis of brightness using Welch's *t*-tests was performed to find the brightest horizon (seaward or landward) using the COA sectors of both the zenith photographs and horizontal photographs as separate measures of horizon. Welch's *t*-tests were also utilised to determine if brightness in the zenith photograph's COA sectors were equivalent to their respective horizontal photograph's COA sectors.

#### 2.4.2. Factors Influencing Brightness on Heron Island

For further analysis of the data a backward stepwise multiple linear regression approach was taken. Models were constructed and analysed based on the differing conditions in which the response variable (brightness) was measured i.e., the photographic direction and sector as well as the moon's presence (Figure 7). In all cases, a primary model was initially constructed in which the response variable was brightness. The explanatory variables included environmental factors (the percentage of the moon illuminated, moon altitude, and cloud cover) which change with time but are expected to vary minimally between adjacent sampling sites due to the relatively small size of the island. Once this primary model was analysed, its residuals were extracted and acted as a proxy for the remaining variation in brightness after the environmental variables had been considered. To understand the remaining variance, the residuals were used as the response variable to a subsequent secondary model in which the explanatory variables were anthropogenic and geographic factors (the number of light sources, percentage of visible sky, percentage cover of buildings, percentage cover of vegetation, and time).

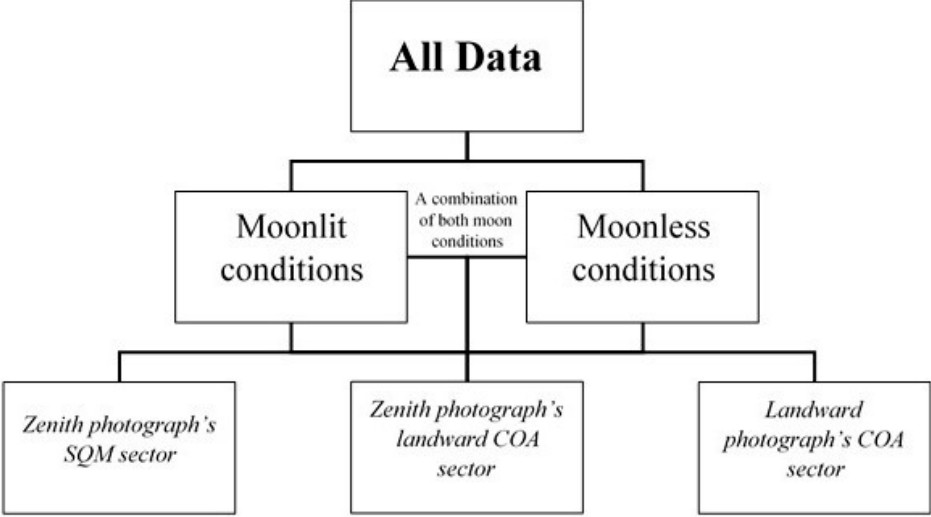

**Figure 7.** Flowchart of the differing conditions (moon presence, photograph type, and sector) in which the response variable (brightness) was measured, factors which dictated the construction of the backward stepwise multiple linear regression models.

### 2.4.3. Factors Influencing Sea Turtle Nesting on Heron Island

A backward stepwise multiple linear regression approach was also taken. The response variable was the combined number of green turtle and loggerhead turtle nests recorded at each sampling site during the 2014–2015 nesting season (more recent data was not available) on Heron Island as adapted from Truscott, Booth & Limpus [26] (Table S3). The explanatory variables included factors likely to influence sea turtle nesting (brightness, rock outcrop presence, average beach width, the number of light sources, the percentage of visible sky, the percentage cover of buildings, and the percentage cover of vegetation). The only variable that differed between models was brightness. For each model, brightness was measured for a specific zenith photograph sector (i.e., SQM sector, landward sector, or the whole photograph), under either moonlit conditions, moonless conditions, or a combination of both. Based on initial analyses, we determined that these models could reliably use zenith photographs for comprehensive readings of brightness at each site without using horizontal photographs (see Section 3.1).

## 3. Results

### 3.1. Spatial Patterns of Brightness on Heron Island

The horizontal and zenith photographs showed no significant difference in their measures of brightness for both the seaward ($p = 0.907$) and landward ($p = 0.096$) COA sectors (Figure 8, Table S4). There was a significant difference between the seaward and landward COA sectors for both the horizontal ($p \le 0.001$) and zenith ($p \le 0.001$) photographs' measures of brightness (Figure 8, Table S4). On average the seaward COA sector was brighter than the landward COA sector regardless of moon presence (Figure 8; Table S5). However, horizontal photographs under moonless conditions, and zenith photographs under a combination of both moon conditions demonstrated a brighter landward direction most notably at sampling sites 3–5 which are located adjacent to the resort (Figures 8 and 9).

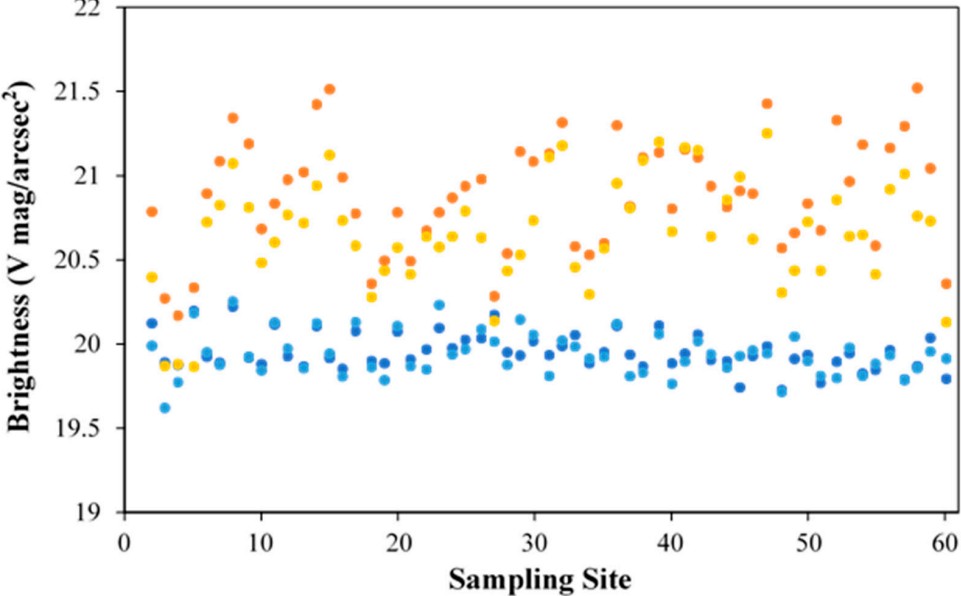

**Figure 8.** Average seaward horizon (shades of blue) and landward horizon (shades of orange) brightness (V mag/arcsec$^2$) at each sampling site as measured by the horizontal photographs' cone of acceptance (COA) sector (dark blue and orange) and zenith photographs' seaward and landward COA sectors (light blue and orange) under a combination of both moon conditions.

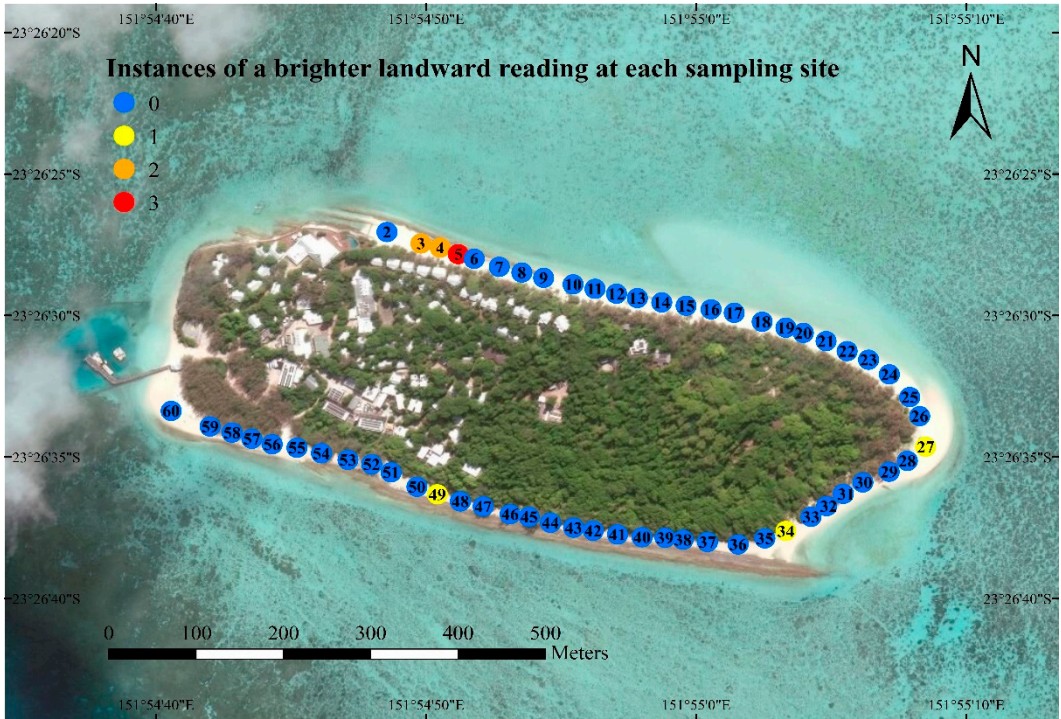

**Figure 9.** Map of Heron Island demonstrating the number of instances in which the landward horizon was on average brighter (V mag/arcsec$^2$) than the seaward horizon, at each sampling site.

### 3.2. Factors Influencing Brightness on Heron Island

The percentage of the moon illuminated was significant for all models in which it was included. Brightness increased (i.e., decreasing Vmag/arcsec$^2$) on average by 0.037 Vmag/arcsec$^2$ per percent increase in the moon illuminated (average $p \leq 0.001$; Tables S6–S9). For all models except one, cloud cover demonstrated a significant trend with brightness which was dependent on the presence of the moon (average $p \leq 0.001$; Tables S6–S9). Whereby, increasing cloud cover resulted in an average increase in brightness of 0.014 Vmag/arcsec$^2$ under moonlit conditions but a decrease in brightness of 0.113 Vmag/arcsec$^2$ under moonless conditions (Figure 10 and Figure S1; Tables S6–S9). Moon altitude was only significant for the models in which brightness was measured using zenith photograph's SQM sector and for horizontal photograph's landward COA sector, in which these models demonstrated an average increase in brightness of 0.011 Vmag/arcsec$^2$ per degree increase in moon altitude (average $p = 0.001$; Tables S6–S9). The environmental factors of these models: the percentage of the moon illuminated, moon altitude, and cloud cover, were able to explain more than 80% of the variation in brightness measured under moonlit conditions, but cloud cover explained less than 40% of the variation in brightness measured under moonless conditions (Figure 11, Tables S6–S9).

Two-thirds of the secondary models significantly demonstrated that an increase in the number of light sources resulted in a 0.007 average increase in the residual brightness (average $p = 0.003$; Figure S2; Tables S6–S9). The percentage of visible sky had a significant interaction with the residual brightness for seven out of the nine secondary models, in which the residual brightness increased on average by 0.031 per percentage increase in visible sky (average $p \leq 0.001$; Figure 12 and Figure S3; Tables S6–S9). These anthropogenic and geographic factors were especially important for explaining up to 40% of the remaining variation in measured brightness under moonless conditions (Figure 13, Tables S6–S9).

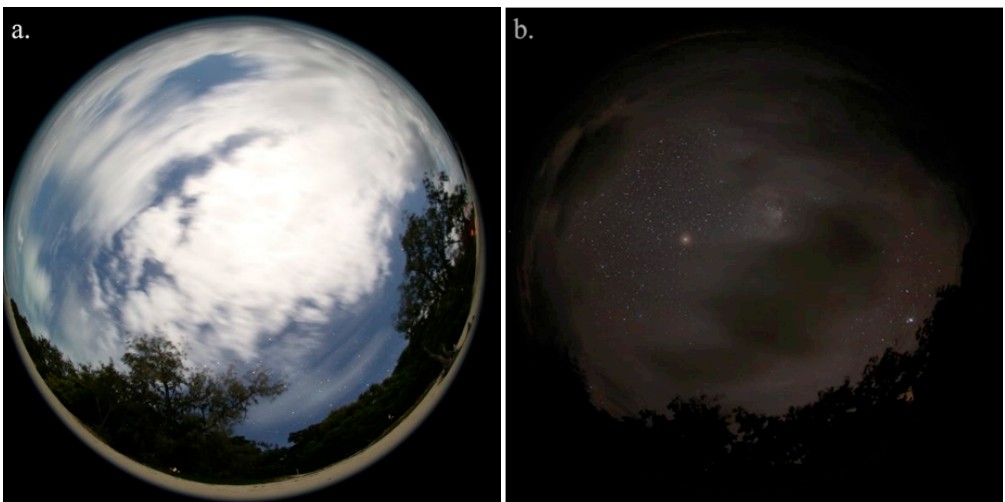

**Figure 10.** Zenith photographs taken on the night of (**a**) 21 June 2018 at sampling site 7 and (**b**) 14 June 2018 at sampling site 23 – Heron Island, Australia. These photographs demonstrate the two states in which clouds can affect brightness: (**a**) increase brightness under moonlit conditions i.e., clouds are brighter than the background sky and (**b**) decrease brightness under moonless conditions i.e., clouds are darker than the background sky.

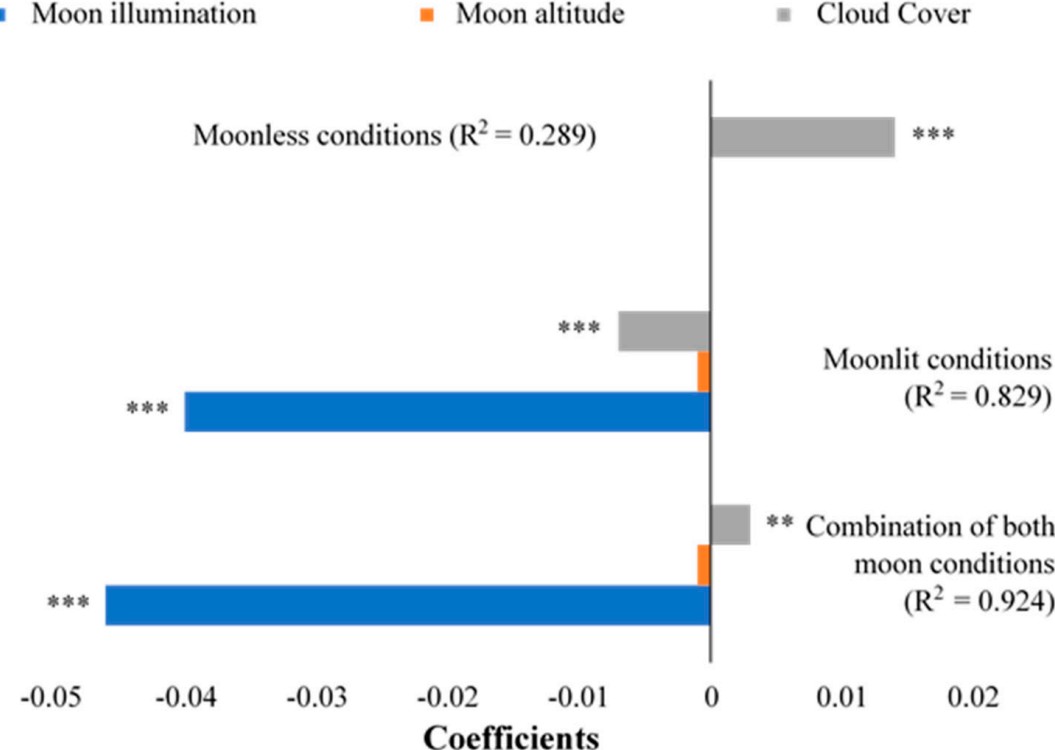

**Figure 11.** The coefficients for the variables determining brightness in the primary backward stepwise multiple linear regression models for zenith photographs' landward cone of acceptance (COA) sectors. Significant variables are denoted with an asterisk (** = $p < 0.01$, *** = $p < 0.001$) and the adjusted $R^2$ for each model shown.

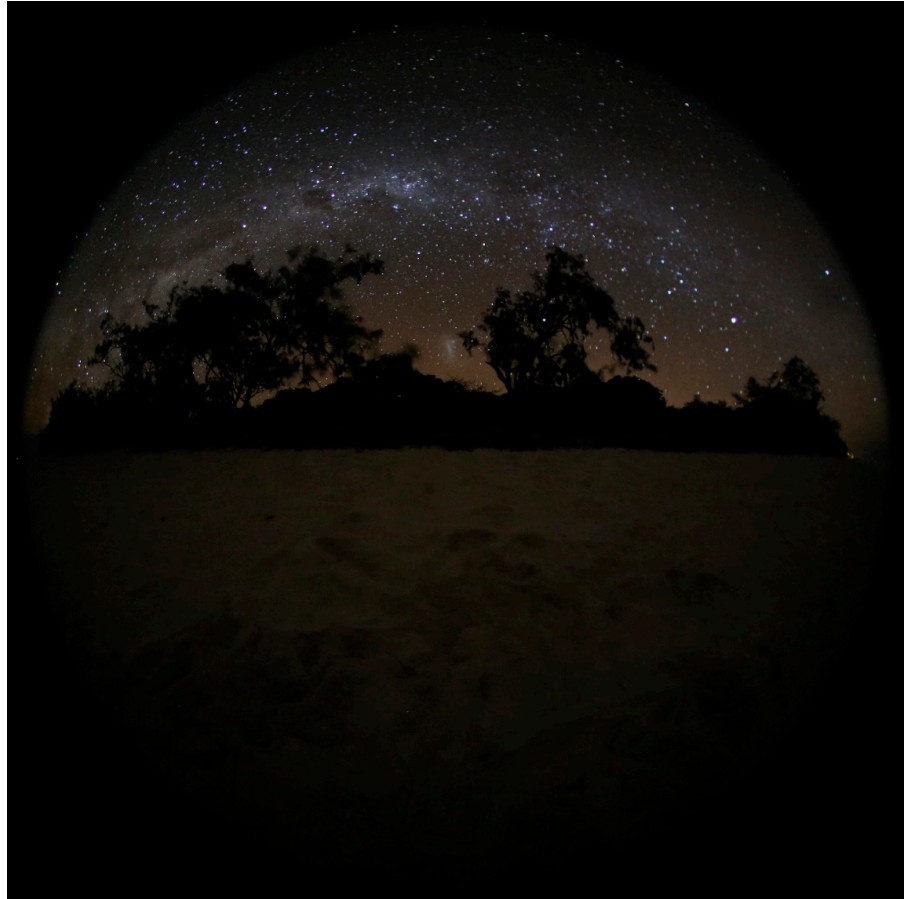

**Figure 12.** Landward photograph taken on the night of 10 May 2018 at sampling site 19 Heron Island, Australia. This photograph clearly demonstrates the contrasting brighter celestial lit sky with the darker vegetative obstruction/silhouettes.

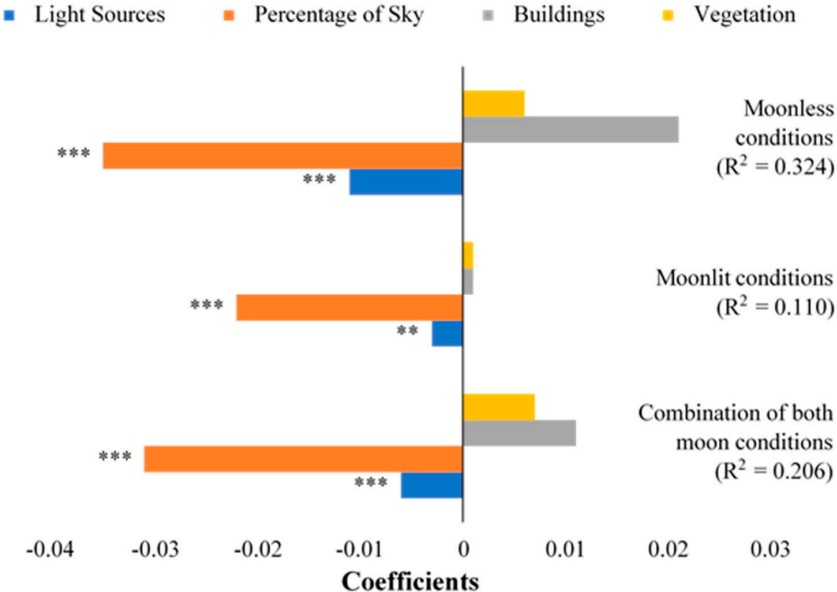

**Figure 13.** The coefficients for the variables determining brightness in the secondary backward stepwise multiple linear regression models for zenith photograph's landward cone of acceptance (COA) sector. Significant variables are denoted with an asterisk (** = $p < 0.01$, *** = $p < 0.001$) and the adjusted $R^2$ for each model shown.

### 3.3. Factors Influencing Sea Turtle Nesting on Heron Island

Overall, the models explained more than 60% of the variation in sea turtle nesting (Figure 14; Table S10). Two variables were significant across all the models explaining sea turtle nesting: the presence of rock outcrop which decreased nesting density on average by 23.563 nests (average $p \leq 0.001$; Figure 14 and Figure S4; Table S10); and the percentage cover of vegetation surrounding sampling sites which decreased nesting density on average by 1.008 nests per percent cover increase (average $p = 0.030$; Figure 14 and Figure S5; Table S10). Only three of the nine models demonstrated significance in other variables, and these models differed in their photographic measures of brightness, being:

1. zenith photographs' landward COA sectors taken under a combination of both moon conditions
2. whole zenith photographs taken under moonless conditions
3. zenith photographs' landward COA sectors taken under moonless conditions

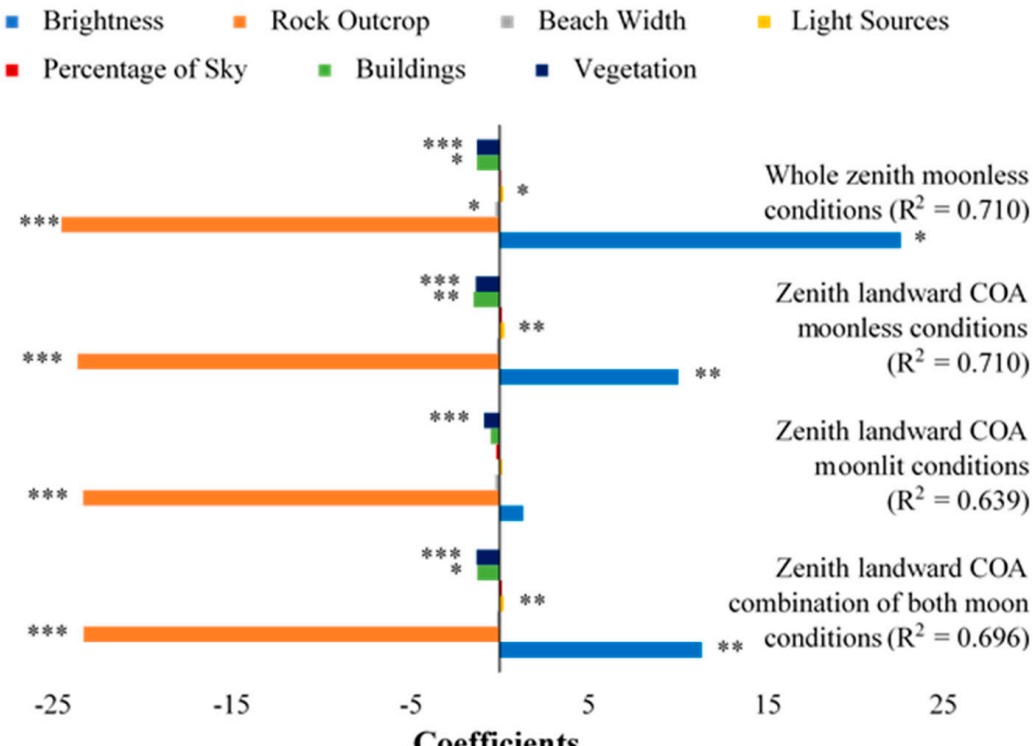

**Figure 14.** The coefficients for the variables determining sea turtle nesting density in the backward stepwise multiple linear regression models for zenith photographs' whole image (top model) and zenith photographs' landward cone of acceptance (COA) sectors (bottom three models). Significant variables are denoted with an asterisk (* = $p < 0.05$, ** = $p < 0.01$, *** = $p < 0.001$) and the adjusted $R^2$ for each model shown.

The above three models significantly demonstrated that on average as brightness increased, nesting density decreased by 14.665 nests (average $p = 0.007$; Figure 14 and Figure S6; Table S10). Furthermore, these three models significantly demonstrated, per percent increase in cover of buildings surrounding sampling sites, nesting density decreased on average by 1.359 nests (average $p = 0.011$; Figure 14 and Figure S7; Table S10). The additional variables, average beach width, the number of light sources, and the percentage of visible sky, had only a very small impact on the overall performance of the models compared to the other more dominant variables (Figure 14).

## 4. Discussion

This study confirmed the commonly held assumption that natural night-time beach brightness in the seaward horizon is brighter than the landward horizon [52–54]. This natural state on Heron Island is likely due to its location away from the mainland (and thus from major light pollution sources), and additional factors including moonlight and presumably starlight, and the reflective nature of the sea collectively causing the seaward horizon to be relatively bright. Conversely, the landward horizon was relatively dark due to vegetative obstruction (Figure 12) [52–54]. However, our findings also suggest that this natural state and the factors controlling it can be altered under conditions of an artificially lit environment, concerningly even at a small local scale as seen on Heron Island. This is of further alarm with increasing coastal urbanisation, particularly in Australia where 85% of the human population already live within 50 km of the coast [55]. We also demonstrated that by using hemispheric night-time imagery, we could quantify and approximate light pollution as experienced by nesting sea turtles, and that this impact was largely dependent on moon presence, moon phase, and cloud cover [23,25,26].

The phenomenon of clouds reflecting artificial light that is emitted upwards has been demonstrated to amplify urban light pollution [21,27–29]. Despite Heron Island not being an urban area, we still expected the same phenomenon to occur and this formed the basis for our hypothesis that cloud cover would result in brighter readings. Instead, our results demonstrated an unexpected relationship between cloud cover and moon presence. Whereby, under moonlit conditions higher percentages of cloud cover resulted in a positive relationship with brightness and under moonless conditions a negative relationship with brightness. The optical depth of clouds determines how moonlight penetrates them and thus ultimately the brightness of the cloud layer as it is observed and measured from the ground beneath [56] (i.e., when the cloud layer is not very thick, it is likely that moonlight can penetrate it and vice versa [21]). Therefore, under moonlit conditions we observed the cloud layer to be brighter than the background sky (Figure 10a). Whereas, under moonless conditions the clouds were not illuminated and were consequentially darker than the background celestial lit sky (Figure 10b) [27,29,41]. These results correspond with previous studies [27,29], which also found that the darkening of night sky brightness by clouds in remote areas (with no or low artificial lighting) is further intensified for clouds at lower altitudes.

Furthermore, previous modelling on the effects of cloud optical depth on the brightness of clouds has demonstrated that within a certain range of cloud optical thickness values (between 1 and 10), the brightness of clouds as observed upwards from the ground, will increase under full moon conditions (similar to the effects found by us, and shown in Figure 10a) [56]. These results from our study and those of others [21,27–29,41,56,57] suggests that there may be a threshold at which artificial light becomes bright enough to effectively compete with moonlight for an interaction with cloud cover. Confirmation of such a threshold will require further investigation into the light being reflected down from, transmitted through, or scattered within clouds and whether the spectral qualities of such night-time brightness corresponds with that of moonlight or artificial light. While Heron Island is a remote island with relatively few lighting sources, clouds were found to both increase and decrease night-time brightness, and their presence should therefore be accounted for when estimating ecological light pollution.

Night-time brightness on Heron Island was relatively site-specific and largely determined by the moon. The percentage of the moon illuminated was the only environmental variable of the primary models that consistently had a significant interaction with brightness—increasing brightness as the moon increased in size. The phenomenon of atmospheric extinction and the radiant extent of moonlight likely explained why the moon's altitude was significantly proportional to brightness—the higher the moon the less atmosphere its light must pass through and the greater its light's radiation per unit area, and thus the brighter the conditions [58]. When considering the moon, the number of sampling sites in which the landward horizon was brighter than the seaward horizon was greater under moonless conditions (six) when compared to moonlit conditions (two; Table S5). Specifically, high brightness in

the landward horizon was recorded (using both photographic directions) under moonless conditions most notably for sampling sites 3 and 4, and 5 (Figures 8 and 9). This was likely due to these sites being located directly adjacent to resort bungalows which projected artificial light unobstructed onto the beach [26]. As expected, these findings establish the moon as the foremost source of night-time light and describe how it mediates the conspicuousness of artificial light/light pollution. However, even though this is not the case world-wide, our methods are still applicable in areas that contrast to Heron Island, such as densely populated areas where high levels of artificial light can overcome the influence of moonlight, thus, entirely altering the natural nightscape [57,59].

Despite the relatively low human activity and presence on Heron Island and the fact that night-time brightness is primarily determined by the moon, we found evidence suggesting that light pollution was still an important factor in sea turtle nest site selection. On Heron Island, the most dominant forces governing sea turtle nesting were rock outcrop presence and the percentage cover of vegetation. The rock outcrops that run parallel to the northern and southern beaches pose major obstacles and thus, sea turtles were three times less likely to nest when a rock outcrop was present (Figure S4) [26]. Along a beachfront a lower percentage cover of vegetation would be expected to lead to a higher percentage cover of sand—making areas where the percentage cover of vegetation is low, high density nesting areas and vice versa, and indeed, our results corroborate this theory. Most notably, our results also demonstrated that high night-time brightness significantly (average $p = 0.007$; Figure 14 and Figure S6; Table S10) decreased sea turtle nesting when brightness was measured under moonless conditions and in the landward direction—spatial and temporal settings in which light pollution is most conspicuous [23–26,60]. The argument for brightness decreasing sea turtle nesting as a function of light pollution is further strengthened by the percentage cover of buildings (a proxy measurement of light pollution) simultaneously and significantly decreasing nesting [17–19]. Furthermore, as previously discussed, our findings described the section of beach encompassing sampling sites 3–5 to have the greatest exposure to artificial light on Heron Island (Figure 9), a state of light pollution exacerbated under moonless conditions [26]. The number of nests at these sampling sites was thus, expectedly significantly fewer than almost all other sampling sites (Table S3), adding additional support to our findings describing the negative impacts of light pollution on sea turtle nesting. Therefore, we suggest that the combination of significance for brightness (as measured in the landward direction and/or under moonless conditions) and the percentage cover of buildings, with sea turtle nesting, demonstrates a light pollution mediated relationship likely resulting in decreased nesting in areas exposed to artificial light.

In this study, we were able to refine methodologies for the use of DSLR cameras with wide angle lenses and the accompanying SQC software, especially regarding ecological applications. Our findings demonstrated that zenith photographs reliably capture equivalent results to horizontal photographs with regards to measuring horizon brightness in the COA sectors of sea turtles. Thus, zenith photography can now be verified to provide comprehensive measurements of brightness in all directions without the need for horizontal photography. However, despite this revelation, whilst zenith photographs were superficially equivalent to horizontal photographs in terms of measuring horizon brightness, the variables which determined this brightness differed between both photographic directions (Tables S6–S9). Consequently, we recommend that future studies should first consider the factors of interest influencing brightness given the scope of the study, i.e., aims, organism and/or ecosystem, and then select the appropriate photographic directions accordingly.

To yield ecologically relevant readings of brightness, measurements must be made within an ecologically relevant zone. In this study, the zone was based on the COA sectors, defined by our model species–sea turtles. Whilst for comparison the SQM sector represented an ecologically irrelevant area and rudimentary measure of brightness. As we expected, comparison of these two methodologies were starkly contrasting. With regards to sea turtle nesting and brightness as a function of light pollution, the SQM sector failed to return any significant/meaningful ecological results (Table S10). Whereas, the COA sectors returned readings of brightness relevant to sea turtles, and thus logical conclusions

could be formed. Recent studies [61,62] have demonstrated that measurements of night-time brightness acquired with hemispheric ground photographs, can be correlated with space-borne night-time imagery. While space-borne remote sensing offers global coverage of night-time lights, the spatial resolution of available global sensors (DMSP/OLS and VIIRS/DNB) is too coarse (3 km and 750 m, respectively; [35]) for detailed studies such as we conducted here (with Heron Island being smaller in size than a single pixel of either of those two sensors). In contrast, commercial night-time sensors such as EROS-B [63] and Jilin-1 [61,64] offer sub-meter spatial resolutions, however, such images are quite expensive (especially if images are to be acquired every night to examine the impact of variations in cloud cover and moon phase), and may not be sensitive enough for scattered low emitting lighting as found on Heron Island. In addition, space-borne sensors cannot measure horizontal light, to which sea turtles (and other organisms) are exposed. Additionally, fieldwork compiling datasets of night-time hemispheric imagery as conducted here, could be complemented in the future by the use of drones. Drones offer both flexibility in their viewing geometry, and in their ability to acquire night-time images at high spatial resolution. Our study therefore, emphasises the importance of using ground-based hemispheric imagery [42] in conjunction with space and air borne sensors to assess ecological light pollution [65], as has been recently recognized in 'The National Light Pollution Guidelines for Wildlife', Commonwealth of Australia 2020 [44].

## 5. Conclusions

This study provides a valuable insight into the spatial and temporal patterns of night-time brightness at a fine scale seldom achieved with regards to the ecological effects of light pollution. The naturally brighter seaward horizon was empirically verified, and we confirmed that the moon has a dominant effect on the natural state of night-time brightness. Moreover, the study demonstrates that this state can be altered by light pollution and its conspicuousness as mediated by the moon. The moon also regulated the influence that cloud cover had on night-time brightness, despite the presence of artificial light. Whilst sea turtle nesting was found to be negatively affected by the conspicuousness of light pollution, the most dominant factors determining nest site selection on Heron Island were rock outcrop presence and the percentage cover of vegetation. More importantly, our model species enabled us to illustrate the advanced capabilities of DSLR cameras with wide angle lenses and the accompanying SQC software. By testing the efficacy of zenith photographs we were able to improve methodological efficiency for future ecological research utilising such an approach, especially with regards to measuring brightness and/or light pollution at the finer scales needed for ecological applications. This contrasts with more rudimentary ground-based remote sensing tools such as SQMs which mostly take point measurements directed upwards, as well as with air and space borne remote sensing techniques which are often limited to measurements taken at lengthy temporal intervals, coarse spatial resolutions, and mostly measure artificial light emitted upwards. Future research should focus on closing the gap between ground based remote sensing and overhead sensors, in order to enable multi-angular remote sensing of night-lights from drones and satellites, to improve mapping of the extent and impacts of light pollution over large areas in general, and for the conservation of sea turtles in particular. These findings were made possible using advanced ground-based remote sensing tools, and our study emphasises the contribution such tools can provide towards advancing ecological applications.

**Supplementary Materials:** The following are available online at: http://www.mdpi.com/2072-4292/12/11/1785/s1.

**Author Contributions:** Conceptualization, J.V., S.K., K.S. and N.L.; Data curation, J.V. and N.L.; Formal analysis, J.V. and N.L.; Funding acquisition, J.V. and S.K.; Investigation, J.V., S.K., K.S. and N.L.; Methodology, J.V., S.K., K.S. and N.L.; Project administration, J.V., S.K. and N.L.; Resources, J.V., S.K. and N.L.; Software, J.V. and N.L.; Supervision, S.K. and N.L.; Validation, J.V., S.K., K.S. and N.L.; Visualization, J.V., S.K., K.S. and N.L.; Writing—original draft, J.V., S.K., K.S. and N.L.; Writing—review & editing, J.V., S.K., K.S. and N.L. All authors have read and agreed to the published version of the manuscript.

**Funding:** This research was funded by The University of Queensland Island Station's Heron Island Research Station Scholarship.

**Acknowledgments:** We would like to acknowledge The University of Queensland's Remote Sensing Research Centre for providing the World View 3 satellite image of Heron Island. We also thank Callum Park for dedicating his time to assisting with fieldwork on Heron Island, and Andrej Mohar for his guidance in using the Sky Quality Camera software.

**Conflicts of Interest:** The authors declare that they have no known competing financial interests or personal relationships that could have appeared to influence the work reported in this paper.

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
