# Peer review of "Quantifying the Impact of Light Pollution on Sea Turtle Nesting Using Ground-Based Imagery"

_remotesensing, doi:10.3390/rs12111785_

Round 1

Reviewer 1 Report

Dear authors,

This is paper is in general well written and focused in a topic that combines measurement techniques of Night Sky Brightness with real application to ecological studies. I think it is a promising work but unfortunately due to some issues with format I cannot review properly sections 2 and 3 so theses issues need to be corrected before a real complete reviewing. 

In any case I have read all the possible sections and there are some comments and suggestions. Specially some of them linked to some references missing specially in the study of clouds effects that are critical for your discussions. Also some clarification in some points and the problem with formats and figures that generates problems to read parts of the manuscript.

I will describe all my comments and suggestions below:

* Abstract 

This abstract is well written and describe quite accurately what will be explained in the manuscript.

Just a ‘typo’ on line 12 where the word Remote appears in bold letters.

* Introduction

This text is very clear too and it describes the state of the art with good references and good approach to the problem the authors want to analyse. Just some few comments around references ti improve a bit this section:

- When authors describe instruments and techniques they are referring to Haenel et al (current citation 22). Maybe it is good for the reader  to indicate this paper is a review of techniques and concludes that the use of DSLR seems one of the best solutions for measuring impact of light.

- When authors describe the effect of clouds and moon on measuring night sky brightness some references are used, but this reference are quite old and only centred in urban locations, so it could be interesting to comment the different effect on natural or polluted areas as it is done in at least this two papers: Ribas et al 2016 on International Journal of Sustainable Lighting and Jechow et al 2017 on Scientific Reports 7. Mainly the sea-turtle study is linked to a non  polluted areas so this clarification of the interaction with clouds is needed (darkening in pristine skies, amplifying in polluted skies) as in fact it is partial said on current line 82 and next ones.

- On line 85 authors describe there is no studies linking effect of clouds with light pollution, the two papers I suggested before are clear examples of their existence and some others specifically linking clouds and ecology like the ones that are current references 23 and 24 by Jechow’s team.

- Finally around the introduction of SQCamera, as far as I know there is no research basic paper on this software but is well described its use in current reference 25 for example, so maybe a short sentence explaining that there is no basic paper but for more details of SQCamera can check this paper of Jechow et al could be a good idea.

* Methods

In general the section seems good focused and just few comments around it. But there is a big issue with the format of figures and some captions and important part of the main text is missed in final part of this section. Some comments:

- There is some alignment problem (Centred) on section 2.1

- Figure 1. It could be good to identify also the buildings cited in the caption of the figure with a another kind of sign in addition to current legend

- On section 2.2, there is the citation of figure A1 from extra materials. I think that is the figure is important to explain the setup conditions needs to appear in main paper. If not maybe it could be simply removed complete because it is properly described in the writing. Personally I propose to add it to the main document.

- Continuing on section 2.2 in equipment there is the description of SQCamera and it refers to paper 24. As I said before it could be good to say this is an example of use (as paper 25 and others) and not a description of SQCamera.

- Caption of Figure 2 has different style of the other ones.

- As far I understand the system is used is a fisheye lens, so in figure 3 it seems that only a portionn of the whole fisheye image is displayed. Maybe it would be more clear to plot the whole circular image or at least indicate that there is a part of the normal fisheye images.

- There is an issue with figures 4 and 5. In fact figure 4 is cut by figure 5 and the caption of fig 4 is missed. Maybe a reduction of size of fig. 4 could be a good idea too.

- Probably for the same issue some text could be missing because I cannot find where figure 5 is cited 

- Figure 6 is almost impossible to check because is partially out of the document.

- Final part of section 2 is broken due to a problem with figures 6 and 7

* Section 3. 

This sections is almost not possible to evaluate because the issue with figures is simply removing title and first subsection. A part of subsection 3.2 is lost too and figure 8 is uncompleted due to format problems So it is recommended to correct this before a real reviewing can be done for this section.

Finally on subsection 3.3 some styling or format issues appear again around lines 308 - 311.

* Discussion

In any case it is a bit complicated to evaluate it without a good reading of section 3 where the core of the research is described. But with the partial reading of section 3 and the reading of section 4 it seems quite clear with just a point that needs to be updated. This is in the discussion around lines 332, where authors discuss around clouds interaction. As I said before in my review there is clear results showing a darkening of the sky in areas with low or no light pollution due to clouds (specially low clouds). A complete analysis of this question is done by Ribas et al 2016 on International Journal of Sustainable Lighting (see https://doi.org/10.26607/ijsl.v18i0.19) and in Jechow et al 2017 on Scientific Reports cited before. I think a reading of both papers is mandatory and can help the discussion, because some open results of this manuscript could be perfectly compatible with those previous works and can consolidate the results and discussion. 

* Conclusions

This section is a good summary of the purposes of this work. Maybe some elements open in discussion that remain for the future could be added to the conclusions section to be because thay can lead future research on this topic.

Reviewer 2 Report

General comments

This is an interesting research about examining the potential for ground photography to quantify the actual light pollution impacting animals. The findings of this study demonstrate the applicability of ground-based remote sensing techniques in accurately and efficiently measuring night-time brightness to enhance our understanding of ecological light pollution. The results of the application of the proposed methodology are very interesting. Furthermore, the case study of sea turtle nesting is very illustrative. However, there are several points in the manuscript that should be clarified. For instance, the economic and time-consuming cost of the method. The cost of the method is higher or lower compared to other remote sensing systems. Or if the same methodology could be applied to other groups of marine animals, or in other ecosystems. Since light pollution also seems to affect other areas in nature.

In general, the manuscript is written correctly, it is very nice and interesting, but the structure could be improved a little more to make it easier to read.

The Introduction section could be rewritten to better order the ideas presented. What is recommended in this section is to move from the most general ideas and cases to the most concrete. And also expose the objectives of the work at the end of the section. In this case it is a bit complicated to follow the speech and they go from arguments without a very clear order.

The Methods and Results sections need to be properly arranged in terms of the position of the figures. As you can download the file in pdf. format from the journal's website, these two sections cannot be read completely, since there are figures that have moved and cover part of the text.

Specific comments

Line 1.- The tittle could result a little bit long. Authors could rewrite the title to shorten it a bit and make it more eye-catching.

Line 12.- The word ‘Remote’ should not be in bold letters.

Line 39.- Keywords should not be repeated in tittle, in order to maximize article visibility in search engines and repositories.

Lines 86, 87.- This paragraph seems to indicate the objective of the work. It is a bit confusing.

Lines 94, 95.- Here, an additional aim of the work is indicated.

Lines 95-97.- The specific main aim is highlighted.

Lines 101-108.- The application of the methodology to two specific examples is mentioned, but perhaps the introductory section could be structured again to expose so many ideas and make the reading more fluid.

Lines 111-117.- The paragraph is centered, pleased justify the text.

Figure 1.- Please include a general location map of the study area. The legend is not seen correctly, as well as geographic coordinates and scale bar please modify these parts of the figure to add more contrast and improve it.

Line 174.- Please unify the text style of the description of Figure 2. Figure 2 presents the same problem in legend and other elements as Figure 1, these zones are not seen correctly.

Line 182.- One of the dots is left over at the end of the sentence.

Line 183.- Measurement of beach features. ‘Figure 3’ should be referenced in this secction of the text.

Line 219.- A part and the description of Figure 4 can not be seen, it appears covered by Figure 5.

Line 224.- ‘Figure 5’ should be referenced in the text.

Line 228.- Two references are needed for both software, Excel and R.

Line 239.- Please indicate the software used for the statistical analysis.

Line 264.- Figure 6 is not placed properly. It appears covering part of the text (the beginning of the Results section) and in addition it is incomplete.

Figure 7 presents the same problem in legend and other elements as previous figures. One of the dots is left over at the end of the sentence.

Lines 276-290.- This paragraph is centered, please justify it.

Figure 8 is not placed properly. It appears covering part of the text.

Lines 308-311.- The font size is bigger, pleased fix it.

Lines 363-364.- In this point of the discussion section an important aspect of the experimental design is pointed out, which does not fit the general cases due to the characteristics of the study area. Please explain this idea better to justify the application of the methodology.

Reviewer 3 Report

From the abstract, title and introduction of the paper, I expected to see some comparison of this DSLR/SQC method and satellite imagery methods and was surprised that comparative analyses were not included in the MS. Showing how the ground base method performs relative to airborne or space based methods would strengthen the impact of this paper. Even if it simply highlights overcoming resolution problems, it would build a stronger case for the ground based method.  

In general, while the MS includes many interesting photographic figures, the layout in the PDF provided in the review had the figures obscuring text and each other. Many of these figures are visually appealing, but some are redundant in the information they provide. Suggest authors consider combining some figures, and making room or adding some data based figures.

Regarding statistical analysis and results, the authors should include the r packages used, and more informative test statistic information than p values only within the text. More information should be included for model evaluation including adjusted R2 values, AIC or other indices, etc. In addition, while having some information in the appendix is fine, I would highly recommend developing a statistics/data based figure or two that correspond to table A5 and some of the other backwards stepwise model comparisons.

Regarding analysis of the turtle nesting data, more emphasis and justification should be included with regard to the time gap between nesting data and brightness data. If the authors have access to other nesting years, they could at least provide an indication of yearly nesting variation, to give the reader a sense of whether the nesting data analyzed represent a typical year.

I recommend restructuring the discussion and conclusion to start with the methodological conclusions (perhaps including discussion of comparison between ground and satellite methods) and end with the ecological conclusions and implications. This is how the rest of the MS is structured and would provide a better flow as well as a better way to show how the methodological advances will improve our knowledge of the ecological systems.

Specific, line-by-line comments are below.

62-63: what is the resolution range? My guess is some remote techniques have resolution that would be at a relevant scale for some ecological studies (depends on species of interest, pollution range etc).

66-68: DSLR with fish eye has been used in ecological studies as an approximation of canopy cover. Might want to re state as there has been at least limited success using this type of technique for canopy coverage, and this paper exemplifies a novel technique for light pollution. Building on previous successes as opposed to inventing the while concept.

66-71: Is there a cost component involved with the accessibility of one method vs another? It would be informative to include that information if one method will save research funds but provide comparative or better data.

69: After line 69, there is no need to spell out Sky Quality Camera, as the abbreviation has already been introduced.

128-129: Does this sentence mean that the camera/lens setup was formatted and calibrated to work with the software? This is what I think the authors intend, but it seems as if they are claiming that the Canon 6D EOS was made to measure light pollution. Just needs a little clarification in wording.

149-152: Were all of the light sources of the same light quality? It seems like number of light sources in a given area would not necessarily indicate light relative light pollution level unless each individual light source gives off the same amount and same types (wavelengths) of light.

175-182: The text does not refer to Figure 3.

195-225: The text refers to this figure as Figure 4, the Figure itself is labelled as Fig 5 (and it should likely be Figure 3 as it is the 3rd non-appendix figure that the text references).

198-225: The way that the PDF was put together, the figures 4&5 are obscuring the main text. I have gone in and moved figures manually, but this may lead to some confusion in reviewer comments….I am not sure if this is an issue with the journal’s formatting, but if not Authors should be aware and fix.

230: when transformations were used, what types of transformations were used and why were they chosen?

Not sure what the dichotomy diagram (all data, moonlit conditions, moonless conditions, etc) refers to as it appears to be cut off and is in front of main text.

303 &306 &313 : too many significant figures. 0.063 0 0.008 of a turtle nest is meaningless…

306: remind us how many models were evaluated here? 3 of how many models demonstrated significance in other variables.

308-311: Upon first reading, I thought number 3 was a subset of number 1. Should clarify this distinction more (that number 1 is combined data). I think that there is a figure the authors tried to include with the dichotomy and perhaps two images, but it is not displaying correctly in the pdf (photos are on a different page and separated from diagram) and there is no legend.

314-318: I found this sentence confusing and contradictory to the previous sentence. If increased light led to decreased nesting density, why would an increase in light source lead to increased nest density? And what does ‘per light source increase’ actually mean?

329-331: Authors did not study sea turtle light perception, and cannot conclude that “We also found that  sea turtles were able to perceive this light pollution…”. They found that measuring the light from a more appropriate angle/scenario to approximate what a turtle might experience allowed for the light pollution to be recorded. Not perception.

Round 2

Reviewer 1 Report

Dear authors,

I think now the paper is almost ready, just very few comments around some sections of the document.

  • Figure 1. For me it is not clear where is the red dot on the Australia situatin map. Maybe it is a problem of my resolution maybe could be more clear.
  • Figure 2 is not appearing on the document.
  • Some captions of figure are at different style than others. I think figure 1 is ok but 3, 4,.. are having problems.
  • Figure 6 is not appearing on the document

I think with this arrangements and formatting adjustments the manuscript could be succesful.

Reviewer 2 Report

I want to thank the authors for responding so kindly to my comments and suggestions.

I think the reviewers' suggestions have helped to improve the manuscript, which was a priori already of high quality.

So I also think it's a pity that some of the figures have a little less quality than others. Specifically, Figures 1, 2 and 9 could be improved with little effort.

In figure 1, the general location of the study area is still not indicated.

In Figures 1, 2 and 9, the texts and legends presented could be changed in color so that they appear a little more clearly.

In addition, Figures 10 and 11 continue overlapped in the body of the text.

Perhaps these little suggestions will serve to improve the overall look of the paper.
